# Ratio of Transmitral E Wave Velocity to Left Atrial Strain as a Useful Predictor of Total and Cardiovascular Mortality in Hemodialysis Patients

**DOI:** 10.3390/jcm9010085

**Published:** 2019-12-29

**Authors:** Wei-Chung Tsai, Wen-Hsien Lee, Pei-Yu Wu, Jiun-Chi Huang, Ying-Chih Chen, Szu-Chia Chen, Po-Chao Hsu, Chee-Siong Lee, Tsung-Hsien Lin, Wen-Chol Voon, Ho-Ming Su

**Affiliations:** 1Graduate Institute of Clinical Medicine, College of Medicine, Kaohsiung Medical University, Kaohsiung 807, Taiwan; azygo91@gmail.com (W.-C.T.); cooky-kmu@yahoo.com.tw (W.-H.L.); karajan77@gmail.com (J.-C.H.); 2Division of Cardiology, Department of Internal Medicine, Kaohsiung Medical University Hospital, Kaohsiung 807, Taiwan; pochao.hsu@gmail.com (P.-C.H.); lcsphk@ms18.hinet.net (C.-S.L.); lth@kmu.edu.tw (T.-H.L.); d750078@kmu.edu.tw (W.-C.V.); 3Faculty of Medicine, College of Medicine, Kaohsiung Medical University, Kaohsiung 807, Taiwan; scarchenone@yahoo.com.tw; 4Department of Internal Medicine, Kaohsiung Municipal Siaogang Hospital, Kaohsiung 812, Taiwan; wpuw17@gmail.com (P.-Y.W.); 990329kmuh@gmail.com (Y.-C.C.); 5Research Center for Environmental Medicine, Kaohsiung Medical University, Kaohsiung 807, Taiwan

**Keywords:** total mortality, cardiovascular mortality, left atrial strain, hemodialysis

## Abstract

The ratio of transmitral E-wave velocity (E) to a left ventricular diastolic parameter is reported to be well correlated with left ventricular filling pressure and is useful in the prediction of mortality. Left atrial (LA) strain has been demonstrated to be associated with left ventricular diastolic function. The aim of the study is to examine the ability of E/LA strain in predicting total and cardiovascular mortality in hemodialysis (HD) patients. In 197 routine HD patients, global LA strain during the reservoir phase was estimated by taking the average of longitudinal strain data obtained from the apical four-chamber and two chamber views by two-dimensional speckle tracking echocardiography. Twenty-nine total mortality and 14 cardiovascular mortality were documented during the 2.7 ± 0.6-year follow-up. After adjusting age, comorbidities, albumin, E/early diastolic mitral annular velocity (Ea), and LA strain, increased E/LA strain (hazard ratio (HR) = 1.191, 95% confidence interval (CI) = 1.072−1.324, *p* = 0.001) was still associated with increased total mortality. After adjusting age, comorbidities, albumin, E/Ea, left ventricular ejection fraction, and LA strain, increased E/LA strain (HR = 1.195, 95% CI = 1.041−1.372, *p* = 0.011) was still associated with increased cardiovascular mortality. In conclusion, E/LA strain is a useful parameter in the prediction of total and cardiovascular mortality in HD patients. Hence, E/LA strain deserves to be calculated in HD patients for better survival prediction.

## 1. Introduction

Cardiovascular disease is the leading cause of mortality in hemodialysis (HD) patients [1]. Calleja et al. found that HD patients had a higher left atrial (LA) volume than normal controls [2]. The left atrium plays an essential role in modulating left ventricular filling, contributing up to a third of cardiac output [3]. Increased LA size has been identified as an important biomarker of left ventricular diastolic dysfunction, cardiovascular disease, and adverse cardiovascular outcomes [4,5,6,7].

Two-dimensional speckle tracking echocardiography (STE) allows accurate assessment of left ventricular systolic and diastolic function. Recently, STE has proved to be useful in measuring LA mechanics. By measuring peak longitudinal LA strain during the reservoir phase, we can obtain the LA reservoir function. LA reservoir function was reported to be able to predict adverse cardiovascular outcomes both in patients with heart failure with preserved [4] and reduced [8] ejection fraction. 

HD patients were found to have a lower LA strain than normal controls [2]. Abid et al. demonstrated that LA strain had a negative correlation with brain natriuretic peptide in HD patients. LA strain has been demonstrated to be associated with left ventricular diastolic function [9,10]. By combining the left ventricular diastolic parameter with transmitral E wave velocity (E), a more accurate estimation of left ventricular filling pressure has been achieved [11,12,13]. The ratio of E to early diastolic mitral annular velocity (Ea) was reported to be well related to mean pulmonary capillary wedge pressure and useful in prediction of mortality [12,14]. Hence, we hypothesize the combination index, E/LA strain, also can predict mortality in HD patients. Hence, the present study is designed to examine the ability of LA strain and E/LA strain in the prediction of total and cardiovascular mortality in HD patients.

## 2. Materials and Methods

### 2.1. Study Population

The study was conducted in a regional hospital in southern Taiwan. All regular HD patients in this hospital were included except those who did not accept echocardiographic examination (*n* = 6), those with severe mitral stenosis or mitral regurgitation (*n* = 3), and those with atrial fibrillation (*n* = 4). Finally, 197 patients were included in this study. 

### 2.2. Ethics Statement

The study protocol was approved by our institutional review board committee of the Kaohsiung Medical University Hospital (KMUH-IRB). Informed consents were acquired from the patients and our study was conducted according to the principles expressed in the Declaration of Helsinki.

### 2.3. Hemodialysis

All patients received regular HD 3 times per week. Every HD session was done for 3–4 h using a dialyzer with a blood flow rate of 250 to 300 mL/min and a dialysate flow of 500 mL/min.

### 2.4. Echocardiographic Measurements

The echocardiographic examination was performed using VIVID 7 (General Electric Medical Systems, Horten, Norway) by an experienced echocardiographer who was blind to the clinical data of the patients, according to standardized protocol. Pulsed tissue Doppler imaging was obtained with the sample volume placed at the lateral and septal corners of the mitral annulus from the apical 4-chamber view. Ea was averaged from the septal and lateral ones. The modified Simpson’s method was used to evaluate the left ventricular ejection fraction (LVEF). Left ventricular mass was calculated using the Devereux-modified method [15]. The left ventricular mass index (LVMI) was calculated by dividing the left ventricular mass by body surface area. The left atrial volume was calculated using the biplane area-length method [16]. The left atrial volume index (LAVI) was calculated by dividing the left atrial volume by body surface area. The average value of these echocardiographic parameters from 3 consecutive cardiac cycles was used for later analysis.

### 2.5. LA Strain Measurement

LA strain was measured from two-dimensional STE. Detail measurement of LA strain was published in our previous study [17]. Global LA strain during the reservoir phase was calculated by taking the average of longitudinal strain data obtained from the apical four-chamber and two chamber views [18,19]. Data acquired from a total of 12 LA segments (annular, mid, and superior segments along the septal, lateral, anterior, and inferior LA walls using apical four-chamber and two-chamber images) were averaged to get global longitudinal LA strain at the end of left ventricular ejection (LA reservoir phase). Assessment of LA strain was accepted if at least 4 of the 6 LA segments in each view could be measured clearly. The raw ultrasonic data was recorded and analyzed offline using EchoPAC software (EchoPAC version 08; GE-Vingmed Ultrasound AS GE Medical Systems).

### 2.6. Collection of Clinical and Laboratory Data

Age, sex, current smoking history, and comorbidities were obtained from medical records or interviews with patients. The body mass index (BMI) was calculated as the ratio of weight in kilograms divided by the square of height in meters. Laboratory data were measured from fasting blood samples and obtained within 1 month of enrollment. Diabetes was diagnosed if fasting blood glucose was ≥126 mg/dL or hypoglycemic agents were used to control blood sugar. Stroke was defined as a history of cerebral bleeding or infarction. Coronary artery disease was defined as a history of typical chest pain with a positive stress test, angiographically documented coronary artery disease, old myocardial infarction, or having undergone coronary artery bypass surgery or angioplasty. Heart failure was defined based on the Framingham criteria [20].

### 2.7. Definition of Cardiovascular Mortality

Cardiovascular mortality was determined and judged by two cardiologists with any disagreement resolved by adjudication from a third cardiologist from the hospital course and medical records. Cardiovascular mortality was defined as death caused by ischemia heart disease, cardiogenic shock, heart failure, lethal arrhythmia, unexplained sudden cardiac death, aortic dissection, and cerebrovascular disease. In mortality patients, they were followed until the date of death. The other patients were followed until March 2017.

### 2.8. Statistical Analysis

We used SPSS 22.0 software (SPSS, Chicago, IL, USA) for statistical analysis. Data were presented as mean ± standard deviation or percentage. Categorical and continuous variables between groups were compared by the chi-square test and independent samples *t*-test, respectively. After we had determined normality using a Kolmogorov–Smirnov test for all continuous variables, appropriate parametric and non-parametric tests were used. We investigated any relationship between two normal distribution variables by Pearson’s correlation method and between two non-normal distribution variables by Spearman’s correlation method. The significant variables in the univariable linear regression analysis were selected for multivariable linear regression analysis. A Kaplan–Meier survival plot was calculated from baseline to time of mortality events and compared using the log-rank test. Time to mortality events was modeled using the Cox proportional hazards model with a forward selection. All tests were 2-sided and the level of significance was established as *p* < 0.05. 

## 3. Results

### 3.1. Baseline Characteristics in Study Patients

Table 1 shows the baseline characteristics in our study patients. Among the 197 subjects, the mean age was 61 ± 12 years, mean LA strain was 22.1 ± 7.8%, and mean E/LA strain was 4.2 ± 2.6 m/s.

### 3.2. Univariable and Multivariable Correlations of LA strain

Table 2 shows the univariable and multivariable correlates of LA strain. In the univariable analysis, decreased LA strain was associated with increased age, the presence of diabetes, coronary artery disease, stroke, and chronic heart failure, and decreased triglyceride and Ea. After multivariable analysis, the presence of diabetes and decreased Ea were still the major determinants of decreased LA strain. 

### 3.3. Univariable and Multivariable Correlations of E/LA strain

Table 3 shows the univariable and multivariable correlates of E/LA strain. In the univariable analysis, increased E/LA strain was associated with increased age, current smoking, the presence of diabetes, coronary artery disease, and chronic heart failure, using of angiotensin converting enzyme inhibitors/angiotensin II receptor blockers, using of β blockers, decreased triglyceride, and increased E/Ea, LAVI, and LVMI. After multivariable analysis, current smoking and increased E/Ea were still the major determinants of increased E/LA strain.

### 3.4. Kaplan–Meier Analyses of Total and Cardiovascular Mortality-free Survival in Our Patients

The follow-up period to mortality was 2.7 ± 0.6 years. Twenty-nine total mortalities and 14 cardiovascular mortalities were recognized during the follow-up period. The causes of mortality included chronic heart failure with acute exacerbation (*n* = 7), acute myocardial infarction (*n* = 3), cerebrovascular disease (*n* = 3), lethal arrhythmia (*n* = 1), malignancy (*n* = 5), infectious disease (*n* = 9), and gastrointestinal bleeding (*n* = 1). 

The best cut-off value of E/LA strain in the prediction of mortality has not been established. To find the appropriate cut-off value of E/LA strain as a predictor of cardiovascular mortality, we created some models using different cut-off values of E/LA strain. Using the chi-square value to choose the model with the best performance, the model using E/LA strain >4.56 m/s had the best performance in predicting cardiovascular mortality. There were 61 patients with E/LA strains of >4.56 m/s. Figure 1 illustrates the Kaplan–Meier curves for total mortality-free survival (Figure 1A) and cardiovascular mortality-free survival (Figure 1B) in study patients subdivided according to E/LA strain >4.56 m/s or not (log-rank *p* ≤ 0.012). Patients with E/LA strain >4.56 m/s had higher total and cardiovascular mortality rates.

### 3.5. Major Predictors of Total and Cardiovascular Mortality in Study Patients

Table 4 shows the predictors of total mortality using the Cox proportional hazards model. In the univariable analysis, increased total mortality was associated with increased age, E/Ea, and E/LA strain, the presence of diabetes, coronary artery disease, and chronic heart failure, and decreased albumin and LA strain. After multivariable analysis, increased age and E/LA strain, the presence of coronary artery disease, and decreased albumin were still the major predictors of increased total mortality. 

Table 5 shows the predictors of cardiovascular mortality using the Cox proportional hazards model. In the univariable analysis, increased cardiovascular mortality was associated with increased age, E/Ea, and E/LA strain, the presence of diabetes and coronary artery disease, and decreased albumin, LVEF, and LA strain. After multivariable analysis, increased age, the presence of coronary artery disease, and increased E/LA strain were still the major predictors of increased cardiovascular mortality. 

## 4. Discussion

This study aimed to evaluate LA strain and E/LA strain in the prediction of total and cardiovascular mortality in HD patients. We found the combination index, E/LA strain, was significantly associated with total and cardiovascular mortality after multivariable adjustment in HD patients. 

Nagy et al. found reduced LA strain but not left ventricular systolic and diastolic parameters were associated with heart failure symptoms and outcome in patients with heart failure with preserved ejection fraction [21]. Malagoli et al. reported LA strain could predict cardiovascular events in patients with chronic heart failure with reduced ejection fraction [22]. In contrast, Modin et al. demonstrated that LA strain is a univariable predictor of cardiovascular morbidity and mortality in the general population. However, LA strain did not remain an independent predictor of outcome after adjustment for clinical parameters [23]. Hence, in patients without heart failure, LA strain might not be a helpful outcome predictor. In the present study, although LA strain was significantly associated with total and cardiovascular mortality in the univariable analysis, the association became insignificant after multivariable adjustment. In HD patients, LA strain might not be a useful parameter in mortality prediction.

Diastolic dysfunction may increase left ventricular filling pressure and is the important mechanism responsible for the pathophysiology of heart failure. By combining the left ventricular diastolic parameters with transmitral E-wave velocity, a more precise estimate of left ventricular filling pressure is achieved. The ratio of transmitral E wave velocity to left ventricular diastolic parameter, such as E/Ea and E/left ventricular early diastolic strain rate, had a significant correlation with left ventricular filling pressure [12,24] and could usefully predict mortality in various populations, including in patients with heart failure with reduced ejection fraction [25], in the general population [25], and in patients with acute myocardial infarction [26,27]. Many previous studies have shown that LA strain has a significant correlation with left ventricular diastolic function [28,29,30]. In the present study, we also found LA strain had a significant correlation with Ea, an established left ventricular diastolic parameter, so the combination index, E/LA strain, should have a significant correlation with left ventricular filling pressure and a potential to predict mortality. In fact, our results showed E/LA strain was highly correlated with E/Ea, a good parameter of left ventricular filling pressure [12] and after adjusting many important clinical and echocardiographic parameters, including E/Ea, LVMI, and LA strain, E/LA strain was still a useful parameter in the prediction of total and cardiovascular mortality in our HD patients. Hence, assessment of the combination index, E/LA strain, is helpful in identifying the high-risk group of increased total and cardiovascular mortality in HD patients.

## 5. Study limitation

There were several limitations to this study. First, the study generality was restricted because we only included study patients from one dialysis clinic in a regional hospital in southern Taiwan. Second, two-dimensional STE can generate LA strain and strain rate curves among different LA cycle phases. We only calculated longitudinal LA strain during the reservoir phase. However, LA strain during the reservoir phase has been considered to be the most well-studied one in predicting mortality among the various LA strain measurements [21,22]. Third, because of the abnormally large left atrium in patients with severe mitral stenosis or mitral regurgitation and the lack of effective atrial contraction in patients with atrial fibrillation, we did not include such patients. Finally, due to the large number of variables in our analysis with only 29 total mortality events and 14 cardiovascular mortality events, the possibility of chance findings and the restricted power should be considered.

## 6. Conclusions

In HD patients, the combination index, E/LA strain, was significantly associated with total and cardiovascular mortality after adjusting several important clinical and echocardiographic parameters. Hence, E/LA strain deserves to be measured in HD patients for better survival prediction. 

## Figures and Tables

**Figure 1 jcm-09-00085-f001:**
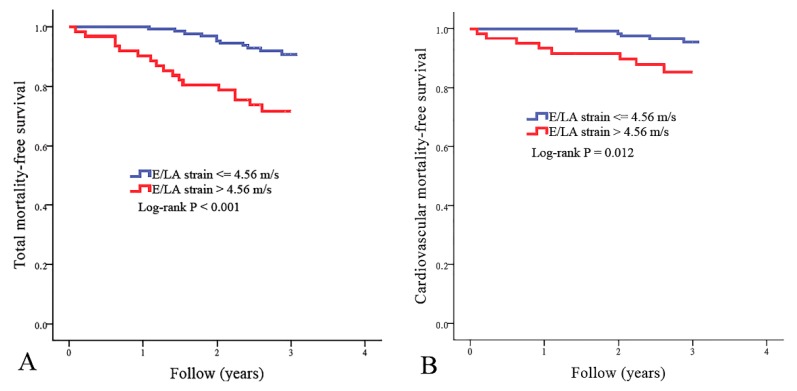
Kaplan–Meier curves for total mortality-free survival ((**A**), log-rank *p* < 0.001) and cardiovascular mortality-free survival ((**B**), log-rank *p* = 0.012) in all study patients subdivided according to the ratio of transmitral E wave velocity (E) to left atrial (LA) strain >4.56 m/s or not.

**Table 1 jcm-09-00085-t001:** Baseline characteristics in our study patients.

Characteristics	All Patients (*n* = 197)
Age (year)	61 ± 12
Male gender (%)	53
Diabetes mellitus (%)	47
Current smoking (%)	14
CAD (%)	10
Stroke (%)	10
CHF (%)	26
SBP (mmHg)	155 ± 27
DBP (mmHg)	82 ± 15
BMI (kg/m^2^)	23.7 ± 3.8
Albumin (g/dL)	3.9 ± 0.3
Hemoglobin (g/dL)	10.5 ± 1.2
Total cholesterol (mg/dL)	178 ± 40
Triglyceride (mg/dL)	167 ± 123
**Medications**	
ACEI and/or ARB use (%)	22
β-blocker use (%)	21
CCB use (%)	25
**Peripheral vascular parameters**	
E (cm/s)	83 ± 30
Ea (cm/s)	6.7 ± 2.4
E/Ea	14.4 ± 9.5
LVEF (%)	62 ± 8
LAVI (mL/m^2^)	35 ± 20
LVMI (g/m^2^)	136 ± 42
LA strain (%)	22.1 ± 7.8
E/LA strain (m/s)	4.2 ± 2.6

ACEI: angiotensin converting enzyme inhibitor; ARB: angiotensin II receptor blocker; BMI: body mass index; CAD: coronary artery disease; CCB: calcium channel blocker; CHF: chronic heart failure; DBP: diastolic blood pressure; E: transmitral E wave velocity; Ea: early diastolic mitral annular velocity; LA: left atrial; LAVI: left atrial volume index; LVEF: left ventricular ejection fraction; LVMI: left ventricular mass index; SBP: systolic blood pressure.

**Table 2 jcm-09-00085-t002:** Univariable and multivariable correlates of LA strain in study patients.

Parameter	Univariable Analysis		Multivariable Analysis	
	*r*	*p*	*β*	*p*
Age (year)	−0.192	0.007	−0.030	0.684
Male gender (%)	−0.066	0.360		
Diabetes mellitus (%)	−0.327	<0.001	−0.183	0.014
Current smoking (%)	−0.120	0.096		
CAD (%)	−0.185	0.010	−0.078	0.260
Stroke (%)	−0.153	0.033	−0.054	0.441
CHF (%)	−0.144	0.045	−0.012	0.865
SBP (mmHg)	−0.051	0.509		
DBP (mmHg)	0.100	0.197		
BMI (kg/m^2^)	−0.134	0.062		
Albumin (g/dL)	0.033	0.650		
Hemoglobin (g/dL)	−0.043	0.553		
Total cholesterol (mg/dL)	−0.008	0.909		
Triglyceride (mg/dL)	−0.179	0.013	−0.059	0.387
**Medications**				
ACEI and/or ARB use (%)	−0.133	0.064		
β−blocker use (%)	−0.106	0.139		
CCB use (%)	−0.091	0.209		
**Echocardiographic parameters**				
E (cm/s)	0.009	0.901		
Ea (cm/s)	0.378	<0.001	0.279	<0.001
LVEF (%)	0.103	0.158		
LAVI (mL/m^2^)	−0.131	0.072		
LVMI (g/m^2^)	−0.117	0.105		

**Table 3 jcm-09-00085-t003:** Univariable and multivariable correlates of E/LA strain in study patients.

Parameter	Univariable Analysis		Multivariable Analysis	
	*r*	*p*	*β*	*p*
Age (year)	0.147	0.044	−0.026	0.716
Male gender	0.108	0.142		
Diabetes mellitus	0.171	0.019	0.046	0.522
Current smoking	0.196	0.007	0.178	0.008
CAD	0.222	0.002	0.113	0.121
Stroke	0.017	0.820		
CHF	0.190	0.009	0.040	0.591
SBP (mmHg)	0.004	0.963		
DBP (mmHg)	−0.103	0.190		
BMI (kg/m^2^)	0.067	0.366		
Albumin (g/dL)	0.026	0.721		
Hemoglobin (g/dL)	−0.022	0.768		
Total cholesterol (mg/dL)	0.014	0.852		
Triglyceride (mg/dL)	0.155	0.035	−0.024	0.727
**Medications**				
ACEI and/or ARB use	0.206	0.005	0.019	0.796
β-blocker use	0.157	0.032	−0.025	0.759
CCB use (%)	0.123	0.095		
**Echocardiographic parameters**				
E/Ea	0.596	<0.001	0.399	<0.001
LVEF (%)	0.047	0.525		
LAVI (mL/m^2^)	0.304	<0.001	0.133	0.054
LVMI (g/m^2^)	0.183	0.012	0.007	0.922

**Table 4 jcm-09-00085-t004:** Predictors of total mortality using the Cox proportional hazards model.

Parameter	Univariable Analysis		Multivariable Analysis (Forward)	
	HR (95% CI)	*p*	HR (95% CI)	*p*
Age (year)	1.070 (1.034–1.107)	<0.001	1.086 (1.036–1.138)	0.001
Male gender	0.947 (0.457–1.962)	0.883		
Diabetes mellitus	3.275 (1.449–7.339)	0.003		
Current smoking	1.030 (0.358–2.959)	0.957		
CAD	4.392 (1.939–9.950)	<0.001	4.965 (1.963–12.557)	0.001
CHF	2.153 (1.028–4.509)	0.037		
SBP (mmHg)	1.009 (0.994–1.025)	0.250		
DBP (mmHg)	1.004 (0.977–1.032)	0.775		
BMI (kg/m^2^)	0.909 (0.814–1.015)	0.094		
Albumin (g/dL)	0.197 (0.088–0.441)	<0.001	0.276 (0.100–0.764)	0.013
Hemoglobin (g/dL)	0.859 (0.634–1.164)	0.330		
**Medications**				
ACEI and/or ARB use	1.482 (0.655–3.351)	0.342		
β-blocker use	1.263 (0.539–2.958)	0.591		
CCB use	1.895 (0.895–4.013)	0.089		
**Echocardiographic parameters**				
E/Ea	1.028 (1.005–1.053)	0.015		
LVEF (%)	0.970 (0.932–1.009)	0.128		
LAVI (mL/m^2^)	1.005 (0.995–1.016)	0.318		
LA strain (%)	0.929 (0.881–0.979)	0.006		
E/LA strain (m/s)	1.167 (1.072–1.171)	<0.001	1.191 (1.072–1.324)	0.001

HR, hazard ratio; CI, confidence interval.

**Table 5 jcm-09-00085-t005:** Predictors of cardiovascular mortality using the Cox proportional hazards model.

Parameter	Univariable Analysis		Multivariable Analysis (Forward)	
	HR (95% CI)	*p*	HR (95% CI)	*p*
Age (year)	1.059 (1.007–1.112)	0.025	1.096 (1.028–1.170)	0.005
Male gender	1.598 (0.535–4.768)	0.397		
Diabetes mellitus	4.620 (1.287–16.585)	0.010		
Current smoking	1.746 (0.487–6.260)	0.386		
CAD	6.421 (2.136–19.300)	<0.001	6.403 (1.848–22.182)	0.003
CHF	2.285 (0.793–6.590)	0.116		
SBP (mmHg)	1.017 (0.995–1.039)	0.125		
DBP (mmHg)	1.024 (0.989–1.061)	0.182		
BMI (kg/m^2^)	0.935 (0.802–1.091)	0.397		
Albumin (g/dL)	0.221 (0.069–0.714)	0.013		
Hemoglobin (g/dL)	0.859 (0.634–1.164)	0.330		
**Medications**				
ACEI and/or ARB use	2.199 (0.735–6.586)	0.148		
β-blocker use	1.605 (0.503–5.125)	0.420		
CCB use	1.723 (0.577–5.142)	0.324		
**Echocardiographic parameters**				
E/Ea	1.031 (1.000–1.065)	0.049		
LVEF (%)	0.941 (0.896–0.989)	0.015		
LAVI (mL/m^2^)	1.008 (0.996–1.020)	0.176		
LA strain (%)	0.919 (0.852–0.993)	0.031		
E/LA strain (m/s)	1.202 (1.074–1.345)	0.001	1.195 (1.041–1.372)	0.011

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
