# Peer review of "Ratio of Transmitral E Wave Velocity to Left Atrial Strain as a Useful Predictor of Total and Cardiovascular Mortality in Hemodialysis Patients"

_jcm, 2019, doi:10.3390/jcm9010085_

Round 1

Reviewer 1 Report

I review with interest this paper by Tsai et al. they examine the ability of E/LA strain in predicting total and cardiovascular mortality in 20 hemodialysis (HD) patients. Few comments that need to be addressed before this is aceptable for review.

Abstract: total should be presented before CV mortality. Please clarify adjustments and preset 95% CI. Background will benefit from rewording.

Introduction: CV background should be presented before echocardiographic background presented.

Methods: present rationale for selection and exclusion of patients.

Please provide additional epidemiological details of how mortality was assessed and how cv nature was ascertained: THIS IS A MAJOR LIMITATION.

Please present rationale for variables included in UNIVARIATE ANALYSIS.

KM and Log Rank should be presented before cox modeling. Please explain why non parametric test was not used. Correlations are not described in methods. Regressions are not described in methods.

Author Response

Reviewer1 comments

1. I review with interest this paper by Tsai et al. they examine the ability of E/LA strain in predicting total and cardiovascular mortality in 20 hemodialysis (HD) patients. Few comments that need to be addressed before this is aceptable for review.

 Answer: Thank you.

2. Abstract:total should be presented before CV mortality. Please clarify adjustments and preset 95% CI. Background will benefit from rewording.

 Answer:Total mortality has been presented before CV mortality. Adjustment has been clarified. 95% CI has been presented. Background has been reworded.

# The ratio of transmitral E-wave velocity (E) to a left ventricular diastolic parameter was reported to be correlated well with left ventricular filling pressure and was useful in prediction of mortality. Left atrial (LA) strain has been demonstrated to be associated with left ventricular diastolic function. The aim of the study was to examine the ability of E/LA strain in predicting total and cardiovascular mortality in hemodialysis (HD) patients.(lines 17-21 in revised manuscript)

#Twenty-nine total mortality and 14 cardiovascular mortality were documented during 2.7 ± 0.6 year follow-up. After adjusting age, comorbidities, albumin, total cholesterol, early diastolic mitral annular velocity (Ea), left ventricular mass index (LVMI), and LA strain, increased E/LA strain (hazard ratio [HR] = 1.255, 95% confidence interval [CI] = 1.048-1.502, P = 0.013) was still associated with increased total mortality. After adjusting age, comorbidities, albumin, total cholesterol, Ea, left ventricular ejection fraction, LVMI, and LA strain, increased E/LA strain (HR = 1.429, 95% CI = 1.087-1.878, P = 0.010) was still associated with increased cardiovascular mortality.(lines 24-31 in revised manuscript)

3. Introduction: CV background should be presented before echocardiographic background presented.

Answer:CV backgroundhas been presented before echocardiographic background.

#Cardiovascular disease is the leading cause of mortality in hemodialysis (HD) patients [1]. Calleja et al. found HD patients had a higher left atrial (LA) volume than normal controls [2]. The left atrium plays an essential role in modulating left ventricular filling, contributing up to a third of cardiac output [3]. Increased LA size has been identified as an important biomarker of left ventricular diastolic dysfunction, cardiovascular disease, and adverse cardiovascular outcomes [4-7].(lines 36-41 in revised manuscript)

4. Methods: present rationale for selection and exclusion of patients.

 Answer: We have added the rationale for selection and exclusion of patients.

#The study was conducted in a regional hospital in southern Taiwan. All regular HD patients in this hospital were included except those who did not accept echocardiographic examination (n = 6), those with severe mitral stenosis or mitral regurgitation (n = 3), and those with atrial fibrillation (n = 4). Finally, 197 patients were included in this study.(lines 60-63in revised manuscript)

#Third, because of abnormally large left atrium in patients with severe mitral stenosis or mitral regurgitation and lacking of effective atrial contraction in patients with atrial fibrillation, we did not include such patients.(lines 227-229in revised manuscript)

5. Please provide additional epidemiological details of how mortality was assessed and how cv nature was ascertained: THIS IS A MAJOR LIMITATION.

Answer: We have provided how mortality was assessed and how CV nature was ascertained.

#Cardiovascular mortality were determined and judged by two cardiologists with disagreement resolved by adjudication from a third cardiologist from the hospital course and medical record. Cardiovascular mortality were defined as death caused by ischemia heart disease, cardiogenic shock, heart failure, lethal arrhythmia, unexplained sudden cardiac death, aortic dissection, and cerebrovascular disease.(lines 106-110in revised manuscript)

#Twenty-nine total mortality and 14 cardiovascular mortality were recognized during the follow-up period. The causes of mortality included chronic heart failure with acute exacerbation (n = 7), acute myocardial infarction (n = 3), cerebrovascular disease (n = 3), lethal arrhythmia (n = 1), malignancy (n = 5), infectious disease (n = 9), and gastrointestinal bleeding (n = 1).(lines 153-157in revised manuscript)

6. Please present rationale for variables included in UNIVARIATE ANALYSIS.

 Answer: We selected important clinical parameters, such as age, gender, comorbidities, blood pressures, body mass index, hemoglobin, albumin, lipid profile, and hypertension medication and important echocardiographic parameters, including E/Ea, LVEF, LAVI, LVMI, LA strain, and E/LA strainin our univariate analysis. These above parameters may have an impact on the mortality prediction.

7. KM and Log Rank should be presented before cox modeling. Please explain why non parametric test was not used. Correlations are not described in methods. Regressions are not described in methods.

 Answer: KM and Log Rank has been presented before Cox modeling. We have used non parametric test in the non-normal distribution variables. Correlations and regressions have been described in methods.

#Kaplan-Meier survival plot was calculated from baseline to time of mortality events and compared using the log-rank test. Time to mortality events wasmodeled using the Cox proportional hazards model with forward selection.(lines 121-123in revised manuscript)

#After we had determined normality using a Kolmogorov–Smirnov test for all continuous variables, appropriate parametric and non-parametric tests were used. We investigated any relationship between two normal distribution variables by Pearson’s correlation method and between two non-normal distribution variables by Spearman’s correlation method. The significant variables in the univariable linear regression analysis were selected for multivariable linear regression analysis.(lines 116-120in revised manuscript)

Reviewer 2 Report

In the manuscript entitled “Ratio of transmitral E wave velocity to left atrial strain as a useful predictor of total and cardiovascular mortality in hemodialysis patients”, the authors investigated the prognostic impact of E wave velocity/left atrial strain (E/LA strain) in hemodialysis (HD) patients. They reported that increased E/LA strain was associated with increased mortality. Thus, they concluded that E/LA strain is a useful parameter predicting total and cardiovascular mortality in HD patients. This challenge is scientifically interesting. However, there are some major concerns to be solved.

#1 The authors have shown that E / LA strain is associated with prognosis in HD patients. However, the pathophysiological significance of this E / LA strain index has not been shown. What does the E / LA strain represent or reflect in terms of cardiac physiology?

#2 The authors focused on parameters related to LA. However, data and considerations regarding patients with atrial fibrillation are lacking. In addition, no data and discussions are given regarding mitral regurgitation.

#3 The multivariate analysis in this study included 12 factors for all 29 deaths. In addition, 12 factors have been introduced for 14 cardiovascular deats. There is concern over statistical overfitting. Furthermore, items that are concerned about multicollinearity are used in multivariate analysis.

#4 The authors should provide a clear breakdown of death. There are various causes of death in HD patients. In particular, the causes of 15 non-cardiac deaths are noteworthy.

#5 (Page 2, Line 77) echocardigraphic > echocardiographic

Author Response

Reviewer 2 comments

In the manuscript entitled “Ratio of transmitral E wave velocity to left atrial strain as a useful predictor of total and cardiovascular mortality in hemodialysis patients”, the authors investigated the prognostic impact of E wave velocity/left atrial strain (E/LA strain) in hemodialysis (HD) patients. They reported that increased E/LA strain was associated with increased mortality. Thus, they concluded that E/LA strain is a useful parameter predicting total and cardiovascular mortality in HD patients. This challenge is scientifically interesting. However, there are some major concerns to be solved.

Answer: Thank you.

The authors have shown that E / LA strain is associated with prognosis in HD patients. However, the pathophysiological significance of this E / LA strain index has not been shown. What does the E / LA strain represent or reflect in terms of cardiac physiology?

Answer: We have added the pathophysiological significance of E/LA strain.

# Diastolic dysfunction may increase left ventricular filling pressure and is the important mechanism responsible for the pathophysiology of heart failure. By combining the left ventricular diastolic parameters with transmitral E-wave velocity, a more precise estimate of left ventricular filling pressure has been achieved [11,12]. The ratio of transmitral E wave velocity to left ventricular diastolic parameter, such as E/Ea and E/left ventricular early diastolic strain rate, had a significant correlation with left ventricular filling pressure [12,24] and could usefully predict mortality in various population, including in patients with heart failure with reduced ejection fraction [25], in general population [26], and in patients with acute myocardial infarction [27,28]. Many previous studies showed LA strain had a significant correlation with left ventricular diastolic function [29-31]. In the present study, we also found LA strain had a significant correlation with Ea, an established left ventricular diastolic parameter, so the combination index, E/LA strain, should have a significant correlation with left ventricular filling pressure and a potential to predict mortality. In fact, our results showed E/LA strain was highly correlated with E/Ea, a good parameter of left ventricular filling pressure [12] and after adjusting many important clinical and echocardiographic parameters, including E/Ea, LVMI, and LA strain, E/LA strain was still a useful parameter in prediction of total and cardiovascular mortality in our HD patients. (lines 203-218 in revised manuscript)

The authors focused on parameters related to LA. However, data and considerations regarding patients with atrial fibrillation are lacking. In addition, no data and discussions are given regarding mitral regurgitation

Answer: We have added considerations and discussion regarding patients with atrial fibrillation and mitral regurgitation.

# The study was conducted in a regional hospital in southern Taiwan. All regular HD patients in this hospital were included except those who did not accept echocardiographic examination (n = 6), those with severe mitral stenosis or mitral regurgitation (n = 3), and those with atrial fibrillation (n = 4). Finally, 197 patients were included in this study. (lines 60-63 in revised manuscript)

# Third, because of abnormally large left atrium in patients with severe mitral stenosis or mitral regurgitation and lacking of effective atrial contraction in patients with atrial fibrillation, we did not include such patients. (lines 227-229 in revised manuscript)

The multivariate analysis in this study included 12 factors for all 29 deaths. In addition, 12 factors have been introduced for 14 cardiovascular deats. There is concern over statistical overfitting. Furthermore, items that are concerned about multicollinearity are used in multivariate analysis.

Answer: We do not include Ea in the revised manuscript due to possible collinearity with E/Ea. Statistical overfitting is a real limitation in the present study. We have added it in the study limitation.

# Finally, due to the large number of variables in our analysis with only 29 total mortality events and 14 cardiovascular mortality events, the possibility of chance finding and the restricted power should be considered. (lines 229-231 in revised manuscript)

The authors should provide a clear breakdown of death. There are various causes of death in HD patients. In particular, the causes of 15 non-cardiac deaths are noteworthy.

Answer: We have provided a clear breakdown of death.

# Twenty-nine total mortality and 14 cardiovascular mortality were recognized during the follow-up period. The causes of mortality included chronic heart failure with acute exacerbation (n = 7), acute myocardial infarction (n = 3), cerebrovascular disease (n = 3), lethal arrhythmia (n = 1), malignancy (n = 5), infectious disease (n = 9), and gastrointestinal bleeding (n = 1). (lines 153-157 in revised manuscript)

(Page 2, Line 77) echocardigraphic > echocardiographic

Answer: Thank you. We have corrected this type error.

Round 2

Reviewer 1 Report

I would like to thank the reviewers addressing mine and  other reviewer comments. The new significance and importance of this work has increased and old my queries have been clarified. I am  recommending acceptance for publication.

Reviewer 2 Report

The revised manuscript has been much improved. The authors described overfitting in the analysis as a limitation. However, the items introduced for multivariate analysis have not been reconsidered, and 12 factors remained to be introduced for 29 deaths and 14 cardiovascular deaths. The concerns about overfitting remain. This is because without statistical validity, these results in this study can mislead readers' clinical decisions.
